# Prevention of maternal and neonatal death/infections with a single oral dose of azithromycin in women in labour in low-income and middle-income countries (A-PLUS): a study protocol for a multinational, randomised placebo-controlled clinical trial

► http://dx.doi.org/10.1136/bmjopen-2022-067581

For numbered affiliations see end of article.

**Correspondence to**
Jennifer Hemingway-Foday;
hemingway@rti.org

Jennifer Hemingway-Foday [1], Alan Tita,[2] Elwyn Chomba,[3] Musaku Mwenechanya,[3] Trecious Mweemba,[3] Tracy Nolen,[1] Adrien Lokangaka,[4,5] Antoinette Tshefu Kitoto,[4,5] Gustave Lomendje,[5] Patricia L Hibberd,[6] Archana Patel,[7,8] Prabir Kumar Das,[7] Kunal Kurhe,[7] Shivaprasad S Goudar,[9] Avinash Kavi [9], Mrityunjay Metgud,[9] Sarah Saleem [10], Shiyam S Tikmani,[10] Fabian Esamai,[11] Paul Nyongesa [11], Amos Sagwe,[11] Lester Figueroa,[12] Manolo Mazariegos,[12] Sk Masum Billah [13,14], Rashidul Haque,[14] Md Shahjahan Siraj [14], Robert L Goldenberg,[15] Melissa Bauserman,[16] Carl Bose [16], Edward A Liechty,[17] Osayame A Ekhaguere [17], Nancy F Krebs,[18] Richard Derman,[19] William A Petri,[20] Marion Koso-Thomas,[21] Elizabeth McClure [1], Waldemar A Carlo[2]

## ABSTRACT

**Introduction** Maternal and neonatal infections are among the most frequent causes of maternal and neonatal mortality, and current antibiotic strategies have been ineffective in preventing many of these deaths. A randomised clinical trial conducted in a single site in The Gambia showed that treatment with an oral dose of 2 g azithromycin versus placebo for all women in labour reduced certain maternal and neonatal infections. However, it is unknown if this therapy reduces maternal and neonatal sepsis and mortality. In a large, multinational randomised trial, we will evaluate the impact of azithromycin given in labour to improve maternal and newborn outcomes.

**Methods and analysis** This randomised, placebo-controlled, multicentre clinical trial includes two primary hypotheses, one maternal and one neonatal. The maternal hypothesis is to test whether a single, prophylactic intrapartum oral dose of 2 g azithromycin given to women in labour will reduce maternal death or sepsis. The neonatal hypothesis will test whether this intervention will reduce intrapartum/neonatal death or sepsis. The intervention is a single, prophylactic intrapartum oral dose of 2 g azithromycin, compared with a single intrapartum oral dose of an identical appearing placebo. A total of 34 000 labouring women from 8 research sites in sub-Saharan Africa, South Asia and Latin America will be randomised with a one-to-one ratio to intervention/

placebo. In addition, we will assess antimicrobial resistance in a sample of women and their newborns.

**Ethics and dissemination** The study protocol has been reviewed and ethics approval obtained from all the relevant ethical review boards at each research site. The

## STRENGTHS AND LIMITATIONS OF THIS STUDY

⇒ This study evaluates the use of routine prophylactic azithromycin in women who undergo labour and attempt a vaginal delivery, including those at high risk of infection due to prolonged labour or membrane rupture, which was identified as a top research priority by the WHO and other stakeholders.

⇒ The study evaluates a simple, inexpensive intervention that can be scaled up to reduce the burden of both maternal and neonatal mortality due to infections if successful.

⇒ To optimise generalisability, this study will be carried out in a racially, ethnically and geographically diverse population in seven low-and-middle-income countries.

⇒ In response to global public health concerns about increasing antimicrobial resistance, this study includes an ancillary study to assess antimicrobial resistance and microbiome changes.

⇒ In some locations, azithromycin and other antibiotics are increasingly being used, especially with increasing use of caesarean section, and thus the impact of the outcome may be diminished.

results will be disseminated via peer-reviewed journals and national and international scientific forums.

**Trial registration number** NCT03871491 (https://clinicaltrials.gov/ct2/show/NCT03871491?term=NCT03871491&draw=2&rank=1).

## INTRODUCTION
### Background
Maternal infection during pregnancy and the puerperium, accounting for 10% of maternal deaths, is a top cause of maternal mortality worldwide.[1] Maternal infections increase neonatal sepsis, a leading cause of neonatal death in low-income countries (LICs).[2–5] Neonatal infection accounts for 15% of neonatal mortality worldwide.[2] According to the WHO, maternal and neonatal deaths from infections have remained unchanged or increased, whereas other causes of death have declined.[5 6]

Innovative, effective and scalable interventions are needed to reduce infection-related maternal and neonatal mortality. The evidence backing current WHO guidelines to prevent and treat peripartum infections is limited.[7] Further, current approaches to address neonatal sepsis have had a limited impact and alternative antibiotic regimens suggesting comparable effectiveness have not reduced newborn deaths.[8–11]

### Risk factors for maternal and neonatal infection and sepsis
Caesarean delivery (CD), especially after labour or membrane rupture, is an important risk factor for maternal peripartum infection.[12] Antibiotic prophylaxis for CD is a well-established, effective strategy to reduce infection.[13–16] However, with a prevalence of <5% in LICs, CDs do not account for most maternal sepsis worldwide.[17] Many peripartum infections in low-and-middle-income countries (LMICs) occur in women who deliver vaginally, with women undergoing prolonged labour or membrane rupture at highest risk. Although most pregnancies are not at high-risk for infection due to prolonged labour or membrane rupture, pregnancies where this does occur account for about 50% of maternal and neonatal infections in LMICs.[1] Therefore, identification and management of prolonged labour and/or membrane rupture are critical to reduce maternal sepsis.[7] Thus, the WHO has identified evaluation of routine prophylactic antibiotics for women undergoing vaginal birth or with prolonged labour/membrane rupture as a top research priority.[1]

### Intrapartum azithromycin to prevent maternal and neonatal infections
A novel approach to prevent maternal and neonatal infection is to target organisms that may be frequent pathogens but are not historically treatment targets.[16 18–21] Azithromycin, which is available as a generic agent, has a bimodal half-life of 70 hours in the non-pregnant population. It is commonly used during pregnancy to treat chlamydia, gonorrhoea and other infections.[22] It provides broad coverage against most common pathogens associated with peripartum infections including gram-positive cocci, genital mycoplasmas and *Ureaplasma* infections, and certain gram-negative Bacilli and anaerobes associated with maternal polymicrobial infections and sepsis.[23] In addition, azithromycin also has activity against Group B *streptococcus*, a major cause of neonatal sepsis in developed countries and possibly in LICs.[24 25]

### Evidence from prior studies
In a US randomised clinical trial (RCT), a 50% reduction in maternal peripartum infection resulted from 500 mg intravenous azithromycin added to the standard prophylactic regimen in high-risk women undergoing CD following labour or membrane rupture >4 hours.[18] These results were observed despite universal antibiotics in both treatment groups. Readmission, unscheduled visits, serious adverse events, postpartum fever or subsequent antibiotic treatment were also reduced in the azithromycin group. A related cost-analysis estimated that adjunctive azithromycin saved US$360 per use in high-risk CDs and US$143 per use in scheduled CDs.[26]

A trial in The Gambia suggested the potential for azithromycin prophylaxis to improve maternal and neonatal outcomes.[19] Among 829 participants randomised to 2 g of azithromycin versus placebo before delivery, maternal infections were lower in the azithromycin group, 3.6% vs 9.2%, respectively, (RR 0.40, 95% CI, 0.22 to 0.71, p=0.002). Among newborns, infections were also lower in the azithromycin group (18.1% vs 23.8%; RR 0.76, 95% CI, 0.58 to 0.99; p=0.052). Maternal and neonatal carriage of infectious organisms was lower in the azithromycin group.

### Rationale for a trial
Drawing from these findings, the role of a single oral dose of azithromycin (plus usual care) to prevent maternal death or peripartum sepsis and intrapartum/neonatal death or sepsis appears promising. Of interest is the subpopulation at highest risk for infection due to prolonged labour and/or prolonged membrane rupture. Considering the success with a single 2 g dose in The Gambian trial, which is bioequivalent to the 500 mg intravenous dose used successfully in the US trial, as well as the 40% bioavailability of oral azithromycin, 2 g is considered the most appropriate dose for the proposed intervention. An RCT is the best design to evaluate efficacy and provide evidence for future policy and clinical practice decisions. Such an RCT in a racially, ethnically and geographically diverse population could be accomplished through the established infrastructure of the Global Network for Women's and Children's Health Research (Global Network), a multicountry research network with sites in India, Pakistan, Bangladesh, Guatemala, Kenya, Zambia and

the Democratic Republic of Congo (DRC) and an established data coordinating centre (DCC).[27]

## METHODS AND ANALYSIS

### Study design and intervention

This study is a masked, placebo-controlled, multicentre, RCT. The intervention is a single 2 g dose of oral azithromycin or identical placebo, administered as four 500 mg pills following randomisation. Both groups will also receive the local standard of care during labour, delivery and postpartum. The trial adheres to the Standard Protocol Items: Recommendations for Interventional Trials statement.[28]

### Primary hypothesis

The Azithromycin-Prevention in Labor Use Study (A-PLUS) trial has both a maternal and neonatal primary hypothesis:

▶ Maternal: a single, prophylactic intrapartum oral dose of 2 g azithromycin given to women in labour will reduce *maternal* death or sepsis.
▶ Neonatal: a single, prophylactic intrapartum oral dose of 2 g azithromycin given to women in labour will reduce *intrapartum/neonatal* death or sepsis.

### Study population

Pregnant women labouring in health facilities at eight Global Network sites will be screened by research staff per the eligibility criteria (box 1). The Global Network sites are described in online supplemental appendix 1 of the trial protocol. The secondary population is a cohort of 5500 high-risk women, defined as term and preterm (≥28 weeks) pregnant women with prolonged labour (≥18 hours) or prolonged membrane rupture (≥8 hours). A cohort of mother–infant dyads will also be randomly selected for an ancillary study to assess antimicrobial resistance and microbiome changes. Recruitment took place from September 2020 to August 2022, and data collection for the primary outcome was completed in October 2022. Data collection for the ancillary study of antimicrobial resistance is expected to be complete by August 2023.

### Patient and public involvement

Patients and/or the public were not involved in study design, conduct or reporting.

### Primary outcomes

The primary outcomes are:

▶ Maternal: incidence of maternal death or sepsis within 42 days post delivery in the intervention versus placebo group.
▶ Neonatal: incidence of intrapartum/neonatal death or sepsis with 28 days post delivery in the intervention versus placebo group.

We use the 2017 WHO definition of sepsis which includes a suspicion of infection and the presence of organ dysfunction based on clinical findings.[7]

---

**Box 1  Eligibility criteria**

**Inclusion criteria**
⇒ Pregnant women in labour with gestational age≥28 weeks (by best estimate) who plan to deliver vaginally in a facility.
⇒ Admission to health facility with clear plan for spontaneous or induced vaginal delivery.
⇒ Presence of one or more live fetus confirmed via a fetal heart rate by Doptone prior to randomisation.
⇒ Age≥18 years (minors 14–17 years eligible in countries where married or pregnant minors or their authorised representatives are legally permitted to give consent).
⇒ Provision of written informed consent.*

**Exclusion criteria**
⇒ Non-emancipated minors (as per local regulations).
⇒ Evidence of chorioamnionitis or other infection requiring antibiotic therapy at time of eligibility (women given single prophylactic antibiotics with no plans to continue after delivery will not be excluded).
⇒ Arrhythmia or known history of cardiomyopathy.
⇒ Allergy to azithromycin or other macrolides that is self-reported or documented in the medical record.
⇒ Any use of azithromycin, erythromycin or other macrolide in the 3 days or less prior to randomisation.
⇒ Plan for caesarean delivery prior to enrolment.
⇒ Preterm labour undergoing management with no immediate plan to proceed to delivery.
⇒ Advanced stage of labour (>6 cm or 10 cm cervical dilation per local standards) and pushing or too distressed to understand, confirm or give informed consent regardless of cervical dilation.
⇒ Not capable of giving consent due to other health problems such as obstetric emergencies (eg, antepartum haemorrhage) or mental disorder.
⇒ Any medical condition considered a contraindication per the judgement of site investigators.
⇒ Previous randomisation in the trial.

* A model written informed consent form is available as online supplemental appendix 2 of the trial protocol.

---

*Maternal sepsis* is defined as organ dysfunction resulting from suspected or confirmed infection that occurs post randomisation during labour or the postpartum period. This WHO definition is operationalised as suspected or confirmed infection based on the presence of fever or hypothermia plus one or more signs of mild-to-moderate organ dysfunction including tachycardia, low blood pressure, tachypnea, altered mental status/confusion, reduced urinary output, jaundice or renal failure.[7 29–31] Components of peripartum infection considered in diagnosing suspected or confirmed infection include clinical chorioamnionitis, endometritis, wound infections (perineal or caesarean), abdominal or pelvic abscess, mastitis/breast abscess or infection, pyelonephritis, pneumonia, and other bacterial infections (table 1).

*Neonatal sepsis* is defined as proven or possible serious bacterial infection (PSBI) or pneumonia, or meningitis. PSBI will be determined using WHO criteria of severe chest in-drawing, fever, hypothermia, no movement at all or movement only on stimulation, feeding poorly or not feeding at all and/or convulsions.[32] Clinical and

**Table 1** Specified infections considered for maternal sepsis diagnosis

| Type of infection | Azithromycin-Prevention in Labor Use Study (A-PLUS) trial definition |
|---|---|
| Chorioamnionitis | Fever (>100.4°F/38°C) in addition to one or more of the following: fetal tachycardia≥160 bpm, maternal tachycardia>100 bpm, tender uterus between contractions or purulent/foul smelling discharge from the uterus prior to delivery. |
| Endometritis | Fever (>100.4°F/38°C) in addition to one or more of maternal tachycardia>100 bpm, tender uterine fundus or purulent/foul smelling discharge from the uterus after delivery. |
| Wound infection | Purulent infection (superficial or deep infection including necrotising fasciitis) of a perineal or caesarean wound with or without fever. In the absence of purulence, a wound infection requires presence of fever (>100.4°F/38°C) and at least one of the following signs of local infection: pain or tenderness, swelling, heat or redness around the incision/laceration. |
| Abdominopelvic abscess | Evidence of pus in the abdomen or pelvis noted during open surgery, interventional aspiration or imaging. |
| Pneumonia | Fever (>100.4°F/38°C) and clinical symptoms suggestive of lung infection including cough and/or tachypnea (>24 breaths/min) or radiological confirmation. |
| Pyelonephritis | Fever (>100.4°F/38°C) and one or more of the following: urinalysis/dip suggestive of infection, costovertebral angle tenderness or confirmatory urine culture. |
| Mastitis/breast abscess or infection | Fever (>100.4°F/38°C) and one or more of the following: breast pain, swelling, warmth, redness or purulent drainage. |

laboratory signs of infection will also be considered for diagnosis.

Centralised adjudication of key infection outcomes will be implemented to standardise the results. Reported antibiotic treatment and culture status will also be considered for outcome adjudication.

The individual components of these primary outcomes will be analysed.

### Secondary maternal outcomes

► The primary maternal outcome in a high-risk for infection population.
► Incidence of specific maternal infections (table 1).
► Subsequent maternal antibiotic therapy after randomisation to 42 days postpartum for any reason.
► Time from drug administration until initial discharge after delivery.
► Maternal readmissions and admissions to special care units within 42 days postpartum.
► Maternal unscheduled visit for care.
► Maternal gastrointestinal symptoms (nausea, vomiting, diarrhoea) or other reported side effects.
► Maternal death due to sepsis using the Global Network algorithm for cause of death.[33]

### Secondary neonatal outcomes

► Neonatal deaths due to sepsis and all-cause neonatal death in all participants and in labouring women at high risk of infection.
► Other neonatal infections (eg, eye infection, skin infection, omphalitis, urinary tract infection, respiratory rate ≥60 breaths/minute).
► Neonatal readmissions and admissions to special care units within 42 days postpartum.
► Neonatal initial hospital length of stay, defined as time of delivery until initial discharge.

► Neonatal unscheduled visit for care.
► Neonatal death due to sepsis using the Global Network algorithm for cause of death.[34]
► Incidence of pyloric stenosis within 42 days of delivery.

### Study procedures

All sites will train research staff in standardised study procedures to administer the intervention and assess outcomes. The sequence of study activities is described (figure 1). Study procedures are also provided (table 2). A more detailed schedule of study procedures is available as online supplemental appendix 3 of the trial protocol.

### Site preparation

Prior to the trial, sites conducted an observational pilot study using the RCT's planned infrastructure to characterise current practices and optimise identification of suspected infection. The pilot study data was used to validate estimates of intrapartum deaths, maternal sepsis and neonatal sepsis for the RCT's sample size. Each site also met with local health authorities and conducted community sensitisation to ensure that study procedures were appropriate for the local context and to encourage facility and community-level engagement.

### Screening

Pregnant women admitted for delivery at predefined health facilities will be screened for eligibility by research staff, per harmonised criteria (box 1). If a contraindication to participation is found, the woman will be excluded.

### Consent

Research staff at each site are responsible for obtaining informed consent prior to complete cervical dilation or dilation limit approved by local authorities. Participants who are randomly selected to participate in the ancillary

**Figure 1** Study Flowchart. *If early consent, confirmation of eligibility and reconfirmation of consent required.

study on antimicrobial resistance will also be consented at this time. To minimise burden on the labouring women, sites may obtain initial consent from potential participants during the antenatal period and reconfirm consent at screening. As literacy levels vary, the consent form will be reviewed verbally. All research staff responsible for consent procedures will be trained and certified in the protection of human subjects and the study-specific consent procedures.

## Masking

Both the azithromycin and placebo will be procured from the same manufacturer (Idifarma, Spain). The packaging will be standardised across sites and labelled as: 'azithromycin 2 g or placebo' with the expiration date and a unique identifier. Clinical and research staff, and participants, will be masked to treatment status unless a serious adverse event potentially related to the treatment requires unmasking. Each site's study pharmacist will monitor randomisation, drug supply and safety.

## Randomisation

A computer-generated randomisation scheme, created and known only by the DCC, will use a randomly permuted block design with randomly varied block to achieve a 1:1

randomisation, stratified by site. Each site will receive study drug directly from the manufacturer, which is sequentially prepared per the randomisation scheme in identical packaging. The study drug will be distributed to participating health facilities and dispensed sequentially by study staff.

## Monitoring before discharge (baseline assessment)

Routine postdelivery care will be provided by clinical providers, trained to contact research staff in case of suspected or confirmed infection. Research staff will collect demographic characteristics, key clinical measures and outcomes from randomisation until discharge. Data will be abstracted from medical records or collected directly from participants, as relevant. All data will be directly entered into a secure electronic data capture platform, Research Electronic Data Capture (REDCap). Deidentified data will then be transmitted to the DCC. In addition, participants will be educated on signs and symptoms of infection and encouraged to call research staff with any concerns.

## Monitoring after discharge (postpartum follow-up assessments)

After discharge, research staff will contact participants at the following timepoints:

**Table 2** Schedule of study procedures

| | Antenatal Care visits | During labour/pre delivery | Post delivery/pre discharge | 3-day postpartum (pp) | 7-day pp | 14-day pp | 28-day pp | 42-day pp |
|---|---|---|---|---|---|---|---|---|
| Community sensitisation | X | | | | | | | |
| Screening | | | | | | | | |
| Eligibility confirmation | | X | | | | | | |
| Clinical assessment | | X | | | | | | |
| Consent | | X | | | | | | |
| Randomisation | | X | | | | | | |
| Drug administration | | X | | | | | | |
| Baseline data collection | | | | | | | | |
| Sociodemographic information | | X | X | | | | | |
| Medical history | | X | X | | | | | |
| Labour and delivery (L&D) information | | | X | | | | | |
| Monitoring | | | | | | | | |
| Drug side effects* | | X | X | X | X | | | X |
| Maternal events during L&D | | X | X | | | | | |
| Neonatal events during L&D | | | X | | | | | |
| Maternal infection/sepsis* | | | X | X | X | | | X |
| Neonatal infection/sepsis* | | | X | X | X | | | X |
| Maternal death* | | | X | X | X | | | X |
| Stillbirth or neonatal death<28 days of birth* | | | X | X | X | X | X | X |
| Infant mortality≥28 days of birth* | | | | | | | | X |
| Pyloric stenosis* | | | X | X | X | | | X |
| Other maternal outcomes* | | | X | X | X | | | X |
| Other neonatal outcomes* | | | X | X | X | | | X |
| Unintended medical visits* | | | X | X | X | | | X |
| Serious adverse events* | | | X | X | X | | | X |

*Events that may be reported by participant between scheduled study visits.

► In-person visits at 3-day, 7-day and 42-day postpartum to identify maternal or infant infection, unexpected medical visits, side effects and other study outcomes. If suspected infection, research staff will collect and send relevant specimens to the local microbiology lab for analysis.

► Supplemental phone contacts at 14-day and 28-day postpartum to review maternal and neonatal signs of infection (WHO criteria). If signs are identified, participants will visit a study facility for further assessment. These supplemental contacts will reinforce the participants' ability to self-assess maternal and neonatal infection and improve identification between the 7-day and 42-day visits.

If indicated, records of non-study visits to health providers during the follow-up period will be reviewed to ascertain study outcomes. Providers may be called if clarification is needed. Readmissions and related diagnoses identified during follow-up visits will be validated through medical record review. All data will be entered into REDCap, and deidentified data will be transmitted to the DCC.

### Safety monitoring

Surveillance of maternal side effects including nausea, vomiting and diarrhoea/loose stools, abdominal pain, vaginitis and dizziness potentially associated with azithromycin will be conducted during labour and postpartum. For infants, findings suggestive of pyloric stenosis will be assessed during the follow-up visits. Maternal and neonatal surveillance will also include assessment of unintended medical visits, maternal deaths, stillbirths, neonatal death<28 days of birth and infant death>28 days. Additional maternal and neonatal risks associated with azithromycin use include anaphylaxis, allergic reactions (rash), liver failure and arrhythmias. Although rare, these side effects will be monitored and reported as serious adverse events. All safety outcomes will be reviewed by the data monitoring committee (DMC) appointed the by National Institute of Child Health and Human Development (NICHD).

### Analytical plan

Baseline demographic characteristics and key clinical measures will be compared between the treatment arms

using contingency table approaches for categorical variables and analysis of variance models' continuous variables.

P values presented will be based on two-sided tests unless otherwise specified, adjusted for site. For most analyses, the interaction between treatment and site will be assessed and if significant, also be presented by site. For continuous outcomes, distributional properties will be evaluated and if required, transformations or non-parametric tests employed. Additional details for potential covariate adjustments in secondary analyses or handling violations of analytic method assumptions will be detailed in the statistical analysis plan. Three key populations are of interest for study analyses:

1. The intention-to-treat (ITT) population will be the primary analysis population and will include all women randomised and their infants. Analyses of this cohort will be conducted based on randomised treatment.
2. The high risk for infection subgroup will include all women in the ITT and their infants' meeting criteria for being high risk (ie, prolonged labour and/or rupture of membranes) at randomisation. Analyses of this cohort will be conducted based on randomised treatment.
3. The as-treated population will include all randomised participants that receive any study drug during the study and their infants. Analyses of this cohort will be conducted based on treatment received.

The final determination of analysis population membership will be via a masked data review prior to final study analyses to address any potential anomalous cases that may arise in this large study population (eg, randomisation/treatment of a woman who is discharged prior to delivery due to false labour or unresponsiveness to induction).

## Primary analysis

Incidence of maternal death or sepsis and intrapartum/neonatal death or sepsis will be compared separately between the treatment arms using generalised linear models fit with each binary outcome separately as the outcome measure. Estimates of RR and associated 95% CIs will be reported. The model will include terms for treatment and site. As randomisation occurs at the pregnancy level and approximately 1%–2% of pregnancies are anticipated to be multiple gestations, models for neonatal outcomes will account for correlation among multiples assuming an exchangeable covariance structure. For the two primary outcomes, analyses will be conducted using the ITT population and the p values associated with the treatment term will be used to formally test each of the two primary hypotheses at the alpha=0.05 level.

As secondary analyses of the primary outcomes, assuming an overall treatment effect is observed, the models will be run including region (Africa, Latin America or Asia) and a treatment by region interaction term. If the interaction term has a p<0.1, effects will be reported by region with treatment effect within region tested at the 0.025 level.

Additional exploratory models will also be run including individually: (1) a treatment-by-site interaction term, (2) any other antibiotic use during labour (yes or no) and its interaction with treatment and (3) mode of delivery (caesarean or vaginal) and its interaction with treatment. If the interaction term for any of these models has a p<0.1, then effects will also be reported by the relevant subgroups.

From each final model, estimates of relative risk associated with treatment will be obtained including unadjusted estimates of risk from the primary model as well as estimates of risk adjusted for potential confounders from the secondary analyses.

## Secondary analyses: women at high risk for infection cohort

The major secondary aim is assessing the two primary outcomes (ie, incidence of maternal death or sepsis and incidence of intrapartum/neonatal death or sepsis) in the high-risk cohort. These analyses will also assess if the treatment effect differs between the high-risk subgroup versus non-high-risk women, defined as all women and their infants in the ITT population that delivered prior to meeting high-risk criteria. Specifically, the maternal and neonatal primary outcomes will be modelled including a treatment by risk status interaction term and excluding data from individuals meeting high-risk criteria after randomisation.

## Secondary analyses: other secondary outcomes

Other maternal and neonatal binary outcomes will be analysed using the approaches described for the primary analysis for the ITT population and approaches detailed for the secondary analysis for the high-risk (HR) cohort. A similar process with generalised linear models employing an appropriate link function will be used to analyse the outcomes of maternal and neonatal initial hospital length of stay. Binary safety outcomes will also be analysed using the approaches detailed for the primary analysis. These analyses will be conducted using the as-treated population.

## Sample size for primary outcome

Sample size estimates were generated to evaluate the potential benefits of peripartum prophylactic azithromycin in two population cohorts of women (table 3). The first population comprises all women delivering in facilities.

Power calculations for the overall study population were generated for the primary maternal and neonatal outcome measures. For each, estimates of the required sample size needed to detect a risk reduction of 20%, 25% and 30% were generated for power of 0.8, 0.85 and 0.9. The risk of sepsis or maternal death was assumed to be 3%, based on data collected from 2010 to 2018 through the Global

**Table 3** (A) Sample size for the overall population (alpha=0.05). (B) Sample size for high-risk group (alpha=0.05). (C) Sample size by region (alpha=0.025)

| Baseline risk | Risk reduction | Evaluable sample size per arm | | |
|---|---|---|---|---|
| | | Power=0.80 | Power=0.85 | Power=0.90 |
| **(A)** | | | | |
| 3% | 20% | 11455 | 13103 | 15334 |
| 3% | 25% | 7133 | 8159 | 9548 |
| 3% | 30% | 4815 | 5508 | 6446 |
| 8% | 20% | 4096 | 4686 | 5483 |
| 8% | 25% | 2554 | 2921 | 3419 |
| 8% | 30% | 1727 | 1975 | 2311 |
| 14% | 20% | 2204 | 2521 | 2950 |
| 14% | 25% | 1377 | 1575 | 1842 |
| 14% | 30% | 932 | 1066 | 1247 |
| **(B)** | | | | |
| 6% | 20% | 5568 | 6369 | 7453 |
| 6% | 25% | 3470 | 3969 | 4644 |
| 6% | 30% | 2344 | 2681 | 3138 |
| **(C)** | | | | |
| 8% | 20% | 4961 | 5607 | 6477 |
| 8% | 25% | 3093 | 3496 | 4038 |
| 8% | 30% | 2091 | 2363 | 2729 |
| 14% | 20% | 2669 | 3017 | 3485 |
| 14% | 25% | 1677 | 1884 | 2176 |
| 14% | 30% | 1129 | 1276 | 1474 |

Network's Maternal Newborn Health Registry, a prospective, population-based registry of pregnant women and neonates receiving care in defined Global Network catchment areas,[27] augmented with active surveillance. For the neonatal outcome, the underlying risk of the combined outcome was estimated to be between 8% and 14% based on Global Network data.[35] We assumed that the risk of sepsis not resulting in death is approximately 4%–10%. We will also test the neonatal risk separately in south Asia and sub-Saharan Africa. Approximately 37.5% of randomised mothers will be from sub-Saharan Africa and 50% will be from Asia, reflecting the rates observed in the Global Network.[35 36]

For the primary neonatal outcome of interest of intrapartum/neonatal sepsis or death, assuming that the loss to follow-up will be in the 2%–3% range, this sample size of 34 000 will provide 90% power to detect a 25% reduction in neonatal mortality and sepsis in the sub-Saharan African region and will provide 90% power to detect a 20% reduction in Asia assuming the baseline risk is at least 8%. For the primary maternal outcome of maternal death or sepsis, the sample size will provide 90% power to detect a 20% reduction from 3% in the population aggregated across all study sites. Each site will aim to recruit an equal number of participants (n=4250 per site) but site-specific sample sizes may be adjusted if targets are not met. No site will be permitted to recruit more than 20% of the overall study sample site.

### Data monitoring plan and stopping rules
Each site will report data, including adverse events, to the Global Network's DCC. The data will be used to evaluate protocol adherence and site performance (eg, recruitment, loss to follow-up, data quality). The DCC will provide standardised progress reports to NICHD and the site investigators monthly to monitor outcome variables and adverse events.

Trial oversight will be handled by two principal groups: (1) a protocol-focused steering committee (table 4) and (2) a DMC, designated by NICHD to ensure safe and ethical treatment of participants through biannual (at minimum) review of data on participant safety, study progress and futility. One formal interim analysis of efficacy and futility will be conducted by the DMC during the study.

### Adverse event monitoring plan
Adverse events will be reported and submitted to the DCC (and IRBs) who will report these cumulative masked data to the DMC in the biannual safety reviews. Safety reports will be reviewed internally by the DCC quarterly and the DMC chair will be notified if any potential safety signals

**Table 4** Azithromycin-Prevention in Labor Use Study (A-PLUS) steering committee

| A-PLUS central study team | |
|---|---|
| **A-PLUS lead study site (University of Alabama–Birmingham/University Teaching Hospital, Lusaka, Zambia)** | |
| Waldemar A Carlo, MD<br>Principal investigator<br>University of Alabama at Birmingham, USA | Alan Tita, MD, PhD<br>A-PLUS lead investigator<br>University of Alabama at Birmingham, USA |
| Elwyn Chomba, MBChB, DCH, MRCP<br>Senior foreign investigator<br>University Teaching Hospital, Lusaka, Zambia | Musaku Mwenechanya<br>Country coordinator<br>University Teaching Hospital, Lusaka, Zambia |
| Trecious Mweemba<br>A-PLUS coordinator<br>University Teaching Hospital, Lusaka, Zambia | |
| **RTI International (Global Network Data Coordinating Center)** | |
| Elizabeth M McClure, PhD<br>Principal investigator<br>RTI International | Tracy Nolen, DrPh<br>Coprincipal investigator and senior statistician<br>RTI International |
| Jennifer J Hemingway-Foday, MPH, MSW<br>A-PLUS protocol manager<br>RTI Internationa | |
| **Eunice Kennedy Shriver National Institute of Child Health and Human Development** | |
| Marion Koso-Thomas, MD, MPH<br>Medical officer<br>Global Network for Women's and Children's Health | |
| A-PLUS Research sites | |
| **Democratic Republic of Congo (Kinshasa School of Public Health/University of North Carolina)** | |
| Antoinette Tshefu, MD, PhD, MPH<br>Senior foreign investigator<br>Kinshasa School of Public Health, Kinshasa, Democratic Republic of Congo | Carl L Bose, MD<br>Principal investigator<br>University of North Carolina School of Medicine, USA |
| Adrien Lokangaka, MD, MPH<br>Country coordinator<br>Kinshasa School of Public Health | Gustave Lomendje<br>A-PLUS coordinator<br>Kinshasa School of Public Health |
| **Guatemala (Instituto de Nutricion de Centro America y Panama (INCAP)/University of Colorado)** | |
| Manolo Mazariegos, MD, MPH<br>Senior foreign investigator<br>INCAP, Guatemala City, Guatemala | Nancy F Krebs, MD<br>Principal investigator<br>University of Colorado Health Science Center |
| Lester Figueroa<br>Country coordinator<br>INCAP, Guatemala City, Guatemala | |
| **Bangladesh (International Centre for Diarrhoeal Disease Research(ICDDR,b)/University of Virginia)** | |
| Rashidul Haque, MD<br>Senior foreign investigator<br>ICDDR,b | William Petri, MD<br>Principal investigator<br>University of Virginia |
| Sk Masum Billah<br>Country coordinator<br>ICDDR,b | Md Shahjahan Siraj<br>A-PLUS coordinator<br>ICDDR,b |
| **Belagavi, India (KLE University's JN Medical College/Thomas Jefferson University)** | |
| Shivaprasad S Goudar MD, MHPE<br>Senior foreign investigator<br>KLE University's J N Medical College | Richard Derman, MD, MPH<br>Principal investigator<br>Thomas Jefferson University |
| Avinash Kavi<br>A-PLUS coordinator<br>KLE University's J N Medical College | Mrityunjay Metgud<br>A-PLUS coordinator<br>KLE University's J N Medical College |
| **Pakistan (Aga Khan University/Columbia University)** | |
| Sarah Saleem, MD<br>Senior foreign investigator<br>Aga Khan University | Robert L Goldenberg, MD<br>Principal investigator<br>Columbia University |
| Shiyam Sunder Tikmani<br>A-PLUS coordinator<br>Aga Khan University | |

**Table 4** Continued

| A-PLUS Research sites | |
| --- | --- |
| Nagpur, India (Lata Medical Research Foundation/Boston University) | |
| Archana Patel, MD, DNB, MSCE<br>Senior foreign investigator<br>Lata Medical Research Foundation | Patricia L Hibberd, MD, PhD<br>Principal investigator<br>Boston University School of Public Health |
| Prabir Das<br>Country coordinator<br>Lata Medical Research Foundation | Kunal Kurhe<br>A-PLUS coordinator<br>Lata Medical Research Foundation |
| Kenya (Moi University School of Medicine/University of Indiana School of Medicine) | |
| Fabian Esamai, MBChB, MMed, PhD<br>Senior foreign investigator<br>Moi University School of Medicine, Eldoret, Kenya | Edward A Liechty, MD<br>Principal investigator<br>Indiana University School of Medicine |
| Paul Nyongesa<br>Country coordinator<br>Moi University School of Medicine, Eldoret, Kenya | Amos Sagwe<br>A-PLUS coordinator<br>Moi University School of Medicine, Eldoret, Kenya |
| Osayame Austine Ekhaguere<br>Coinvestigator<br>Indiana University School of Medicine | |

are identified to allow for more frequent DMC monitoring if needed.

## ETHICS AND DISSEMINATION
### Ethics approval and consent to participate
The A-PLUS study protocol has been reviewed and approved by the relevant ethics committees and regulatory authorities at each research site, including the institutional review board at the University of Alabama at Birmingham, which serves as the lead site, and the DCC at RTI International (see online supplemental file 1 for current V.1.6, dated 13 July 2022). The DMC has reviewed the study protocol and will continue to review throughout the enrollment period. The study was registered (clinicaltrials.gov NCT03871491).

All research staff responsible for obtaining informed consent will be trained and certified in the protection of human subjects and the study-specific consent procedures. A model written informed consent form, developed according to the requirements of the US Office for Human Research Protection (OHRP), can be found as online supplemental appendix 2 of the trial protocol. The model consent may be modified by each site to conform to local standards, but the OHRP required elements must be maintained.

### Potential risks and benefits to participation
There are several potential direct and indirect benefits of this trial. Emerging data suggest that intrapartum azithromycin reduces maternal and neonatal infection. As infections are a frequent cause of maternal and neonatal deaths, there is a possibility that mortality could be reduced at the participating sites, as well as worldwide.

An ongoing concern for peripartum and perinatal antibiotic prophylaxis is the selection of resistant organisms including azithromycin-resistant organisms leading to resistant infections, and concern that disruption of gut and other flora (microbiome) in women and particularly in neonates may lead to adverse events including allergic reactions, rash and childhood asthma.[16 28 37] There is a paucity of data to address concerns that disturbances in the establishment of the indigenous intestinal microbiome caused by antibiotic exposure in early life or CD, either directly or through modifications of breast microbiome, may increase risk of immune-mediated and inflammatory conditions later in life.[38–41] In response to these global public health concerns, this protocol includes an ancillary study to monitor for antimicrobial resistance and changes to the maternal and newborn microbiome.

### Dissemination
Following completion of primary data collection, we will disseminate findings through international meetings and high impact peer-reviewed journals. Per the National Institutes of Health (NIH) and Global Network Data Management and Sharing Policies, final research data will also be made publicly available through the NICHD Data and Specimen Hub system within 1 year of publishing the primary manuscript. All data will be deidentified and shared under the assurance of confidentiality and approval from the relevant IRBs. Trial results will also be shared with community members, health workers, health officials and other stakeholders from the areas where participants were recruited. If the results are positive, we will facilitate practice change at participating sites and approach the WHO to evaluate guideline updates based on study findings.

**Author affiliations**
[1]RTI International, Research Triangle Park, North Carolina, USA

[2]The University of Alabama at Birmingham School of Medicine, Birmingham, Alabama, USA
[3]University of Zambia, University Teaching Hospital, Lusaka, Zambia
[4]University of Kinshasa, Kinshasa, Congo (the Democratic Republic of the)
[5]Kinshasa School of Public Health, Kinshasa, Congo (the Democratic Republic of the)
[6]Boston University School of Public Health, Boston, Massachusetts, USA
[7]Lata Medical Research Foundation, Nagpur, Maharashtra, India
[8]Datta Meghe Institute of Higher Education & Research (Deemed to be University), Wardha, Maharashtra, India
[9]KLE Academy of Higher Education and Research, Jawaharlal Nehru Medical College, Belgavi, Karnataka, India
[10]Community Health Sciences, The Aga Khan University, Karachi, Pakistan
[11]Moi University School of Medicine, Eldoret, Kenya
[12]Instituto de Nutricion de Centroamerica y Panama, Guatemala, Guatemala
[13]The University of Sydney, Sydney, New South Wales, Australia
[14]International Centre for Diarrhoeal Disease Research Bangladesh, Dhaka, Bangladesh
[15]Columbia University, New York, New York, USA
[16]The University of North Carolina at Chapel Hill School of Medicine, Chapel Hill, North Carolina, USA
[17]Indiana University School of Medicine, Indianapolis, Indiana, USA
[18]University of Colorado School of Medicine, Denver, Colorado, USA
[19]Office of Global Affairs, Thomas Jefferson University, Philadelphia, Pennsylvania, USA
[20]University of Virginia, Charlottesville, Virginia, USA
[21]Eunice Kennedy Shriver National Institute of Child Health and Human Development, Bethesda, Maryland, USA

**Acknowledgements** The authors would like to acknowledge the A-PLUS research teams for their many contributions to this work: Kinshasa School of Public Health, Kinshasa, Democratic Republic of Congo: Michel Kalonji and Miyalu Junior; University of North Carolina, Chapel Hill, North Carolina, USA: Jackie Patterson; University Teaching Hospital, Lusaka, Zambia: Ernest Banda, Mwansa Chimfwembe, Ruth Nakazwe and Abigail Mwapule; Instituto de Nutrición de Centroamérica y Panamá, Guatemala City, Guatemala: Maynor Manrique; University of Colorado School of Medicine, Denver, CO, USA: Jamie Westcott; International Centre for Diarrhoeal Disease Research, Dhaka, Bangladesh: Qazi Sadeq-ur-Rahman, Amita Farzana, Farhana Jahan, Zarin Tasnim Maliha and Lolit Singh; University of Virginia, Charlottesville, Virginia, USA: Chris Chisolm; KLE Academy Higher Education and Research, J N Medical College Belagavi, Karnataka, India: Sheetal Harakuni, Manjunath Somannavar and Kadappa Beniwadi; Thomas Jefferson University, Philadelphia, Pennsylvania, USA: Frances Jaeger; Aga Khan University, Karachi, Pakistan: Zaheer Habib, Imran Ahmed, Farnaz Naqvi, Naija Ghanchi and Saleem Jessani; Lata Medical Research Foundation, Nagpur, India: Vaishali Khedikar, Savita Bhargav, Samreen Sadaf and Chaitali Gedam; Moi University School of Medicine, Eldoret, Kenya; Milsort Kemboi, Anderson Misati and Kevin Otieno; Indiana School of Medicine, University of Indiana, Indianapolis, Indiana, USA: Sheri Bucher; RTI International, Research Triangle Park, North Carolina, USA: Anna Aceituno, Kay Jackson, Jean Kim, Janet Moore, Suchita Parepalli, Marissa Trotta and Alexis Williams. Bill & Melinda Gates Foundation—Laura Lamberti.

**Contributors** AT and WAC conceived of the study and developed the protocol, with input from EMC, TN, JH-F, MK-T and PLH. TN developed the statistical analyses plan. JH-F wrote the first draft of the manuscript and subsequent revisions, with critical feedback from WAC, AT, EMC, MK-T and TN. EC, MMwenechanya, TM, AL, ATK, GL, PLH, AP, PKD, KK, SSG, AK, MMetgud, SS, SST, FE, PN, AS, LF, MMazariegos, SMB, RH, MSS, RG, MB, CB, EAL, OAE, NFK, RD and WAP contributed to the refinement and finalisation of the study protocol and trial implementation. All authors contributed to the preparation of this manuscript and have reviewed and approved the final version.

**Funding** The A-PLUS study is supported through institutional grants from the Eunice Kennedy Shriver National Institute of Child Health and Human Development (NICHD) (RTI International (U01 HD040636), University of North Carolina at Chapel Hill (U10 HD076465), University of Alabama at Birmingham (U10 HD078437), University of Colorado (U10 HD076474), Thomas Jefferson University (U10 HD076457), Columbia University (U10 HD078438), Boston University (U10 HD078439), Indiana University (U10 HD076461)) and a grant from the Foundation for the National Institutes of Health (MCCL19APT) through the Maternal, Newborn & Child Health Discovery & Tools initiative of the Bill & Melinda Gates Foundation (BMGF) (INV-008973). The views expressed in this manuscript are those of the authors and do not necessarily represent the views of the NICHD; the National Institutes of Health; or the US Department of Health and Human Services or the BMGF.

**Competing interests** None declared.

**Patient and public involvement** Patients and/or the public were not involved in the design, or conduct, or reporting, or dissemination plans of this research.

**Patient consent for publication** Not applicable.

**Provenance and peer review** Not commissioned; externally peer reviewed.

**ORCID iDs**
Jennifer Hemingway-Foday http://orcid.org/0000-0001-9776-5306
Avinash Kavi http://orcid.org/0000-0002-2176-4697
Sarah Saleem http://orcid.org/0000-0002-6797-8631
Paul Nyongesa http://orcid.org/0000-0002-2896-4720
Sk Masum Billah http://orcid.org/0000-0002-8690-6932
Md Shahjahan Siraj http://orcid.org/0000-0001-6925-324X
Carl Bose http://orcid.org/0000-0002-8727-8640
Osayame A Ekhaguere http://orcid.org/0000-0002-8926-3709
Elizabeth McClure http://orcid.org/0000-0001-8659-5444

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
