## [Reviewer comments · BMJ Open]

ARTICLE DETAILS

TITLE (PROVISIONAL)	revention of maternal and neonatal death/infections with a single oral dose of azithromycin in women in labor in low- and middle-income countries (A-PLUS): Study protocol for a multi-national, randomized placebo-controlled clinical trial
AUTHORS	Hemingway-Foday, Jennifer; Tita, Alan; Chomba, Elwyn; Mwenechanya, Musaku; Mweemba, Trecious; Nolen, Tracy; Lokangaka, Adrien; Tshefu Kitoto, Antoinette; Lomendje, Gustave; Hibberd, Patricia; Patel, Archana; Das, Prabir Kumar; Kurhe, Kunal; Goudar, Shivaprasad S.; Kavi, Avinash; Metgud, Mrityunjay; Saleem, Sarah; Tikmani, Shiyam; Esamai, Fabian; Nyongesa, Paul; Sagwe, Amos; Figueroa, Lester; Mazariegos, Manolo; Billah, Sk Masum; Haque, Rashidul; Shahjahan Siraj, Md; Goldenberg, R; Bauserman, Melissa; Bose, Carl; Liechty, Edward A; Ekhaguere, Osayame; Krebs, Nancy F; Derman, Richard; Petri, William A.; Koso-Thomas, Marion; McClure, Elizabeth; Carlo, Waldemar

VERSION 1 – REVIEW

REVIEWER	Mabey, David Department of Clinical Sciences
REVIEW RETURNED	01-Nov-2022

GENERAL COMMENTS	This is a large multi-centre study addressing an important question, the results of which could have important implications for improving maternal and neonatal outcomes in low and middle income countries. Some sections would benefit from clarification. The primary outcomes are said to be:  • Maternal: Incidence of maternal death or sepsis within 42 days post-delivery in the intervention vs. placebo group. • Neonatal: Incidence of intrapartum/neonatal death or sepsis within 28 days post-delivery in the intervention vs. placebo group. However, it is stated that the primary analysis of both the maternal and neonatal primary outcomes will be run including a treatment by risk status interaction term and exclude data from individuals meeting high-risk criteria after randomization, suggesting that the primary outcome is the incidence of death or sepsis in those at low risk, which would be of less interest. The assumptions underlying the sample size calculations are not well explained. It is stated that the risk of sepsis or maternal death was assumed to be 3% in the population comprising all women delivering in facilities, but no justification is given for this assumption. Is it based on preliminary data from the study sites? For the neonatal outcome, it is stated that the underlying risk of the combined outcome was estimated to be 8% and 14% based on
---

	Global Network data. Why are there two different estimates? Does this mean between 8 and 14% ? Other assumptions shown in table 4 include 8% and 14% “by region”, and they plan to test the neonatal risk separately in south Asia and sub-Saharan Africa , but the sample sizes for South Asia, Africa and Latin America are not provided.
--	---

REVIEWER	Oldenburg, Catherine University of California San Francisco, Francis I Proctor Foundation
REVIEW RETURNED	18-Nov-2022

GENERAL COMMENTS	This report is a trial protocol for an exciting and large randomized controlled trial of a single intrapartum 2 g dose of azithromycin versus placebo for reducing maternal and neonatal death or sepsis. The trial is ongoing, and these comments are meant purely for clarification or additional justification. Congratulations to the authors for undertaking this impressive and important trial.  1. Introduction, Line 51: what does “they” refer to here? I assume high risk for infection pregnancies? 2. Introduction Page 4, Line 40: Could the authors provide justification for the dose beyond “seems appropriate”? The previous efficacy in the trial in The Gambia is good rationale; are there other indications for which a 2 g dose is used? 3. Introduction, Page 4, Line 40: The RCT design is the best design to evaluate efficacy, which provides the best evidence for policy. Perhaps consider rewording? 4. Methods, Page 11, Line 17: Statistical tests for baseline variables are not considered best practice – see for example https://pubmed.ncbi.nlm.nih.gov/1967441/, https://pubmed.ncbi.nlm.nih.gov/2171874/ for rationale 5. Methods, Page 11, Line 37: Although adherence concerns with single dose azithromycin are often minimal, I would urge caution with an as-treated analysis unless a modern method that does not rely on classic “per protocol” analysis is used – e.g., an instrumental variable approach or similar. Conditioning on individuals who complied with their randomized treatment assignment can introduce bias. Again, if only a few individuals do not take their randomized treatment assignment, the bias introduced will be minimal if at all, and this is not a concern. But if it is substantial, consider alternative approaches. 6. Methods, Page 14, Line 5: Why is an alpha of 0.05 used given that there are two co-primary outcomes? Typically, we’d split the P-value for an alpha of 0.025 for each primary outcome. This needs justification. 7. Although only noted for the subgroup analyses, the authors note they will adjust for “demographic or clinical variables found to differ significantly between treatment arms”. Consider revising or justifying this approach further – first, as noted previously, we don’t typically look for statistically significant differences in baseline variables (by definition, differences are due to chance – and with a large sample size such as in this trial, very small differences could be statistically significant). Second, post hoc adjustment that is not pre-specified leaves one open to “fishing”. Adjustment for baseline variables in RCT analyses is certainly acceptable and sometimes preferable (e.g., for baseline variables known to be correlated with the outcome) when pre-specified, but otherwise should be avoided. Given that these are secondary analyses, this is not a major concern, but something for consideration and justification, and any
---

	unplanned (post hoc) adjustments should be carefully labeled as such. For discussions on appropriate adjustment in RCT analyses, see for example: https://pubmed.ncbi.nlm.nih.gov/1651207/, https://pubmed.ncbi.nlm.nih.gov/9620808/ 8. Are there any plans for evaluation of treatment effects or effect modification on the additive scale?
--	---

VERSION 1 – AUTHOR RESPONSE

Reviewer 1 (Prof. David Mabey, Department of Clinical Sciences, London School of HTM)

Comments to the Author:

This is a large multi-centre study addressing an important question, the results of which could have important implications for improving maternal and neonatal outcomes in low- and middle-income countries. Some sections would benefit from clarification. The primary outcomes are said to be:

- Maternal: Incidence of maternal death or sepsis within 42 days post-delivery in the intervention vs. placebo group.
- Neonatal: Incidence of intrapartum/neonatal death or sepsis with 28 days post-delivery in the intervention vs. placebo group.

However, it is stated that the primary analysis of both the maternal and neonatal primary outcomes will be run including a treatment by risk status interaction term and exclude data from individuals meeting high-risk criteria after randomization, suggesting that the primary outcome is the incidence of death or sepsis in those at low risk, which would be of less interest.

RESPONSE: Thank you for providing an opportunity to clarify. The primary analysis is based on the overall treatment effect and does not include risk status interaction term. The inclusion of the risk status interaction term is relevant to the secondary objective of assessing the treatment effect in the high-risk cohort. We mistakenly referred to the primary analysis in the “Secondary Analyses: Women at high-risk for infection cohort” section of the manuscript. This reference has been removed.

The assumptions underlying the sample size calculations are not well explained. It is stated that the risk of sepsis or maternal death was assumed to be 3% in the population comprising all women delivering in facilities, but no justification is given for this assumption. Is it based on preliminary data from the study sites?

RESPONSE: The assumed risk of maternal sepsis or death is based on data collected from 2010-2018 for the Global Network’s Maternal Newborn Health Registry (MNHR), a prospective, population-based registry of pregnant women and neonates receiving care in defined catchment areas of the Global Network sites. Although this is slightly higher than the ~2% current risk observed in the MNHR data, we anticipate that with active surveillance rather than passive reporting based on the 2017 WHO definition of maternal sepsis (designed to catch more cases of sepsis), the risk will be at least 3%. We have added this clarifying text to the “Sample Size for Primary Outcome” section of the manuscript. For the neonatal outcome, it is stated that the underlying risk of the combined outcome was estimated to be 8% and 14% based on Global Network data. Why are there two different estimates? Does this mean between 8 and 14%? Other assumptions shown in table 4 include 8% and 14% “by region”, and they plan to test the neonatal risk separately in south Asia and sub-Saharan Africa, but the sample sizes for South Asia, Africa and Latin America are not provided.

RESPONSE: Thank you for the opportunity to provide further clarification. Please see below.

- The underlying risk of the combined outcome is between 8-14%. We have added this clarification to the manuscript text. These estimates are based on data collected from our on-going Maternal and Newborn Health Registry (MNHR). The MNHR does not collect infection status as we do for the A-PLUS trial; therefore, we assumed a wide range in our underlying risk assumption.
- The region-specific power estimates are based on the overall sample size, as described in Table 4 and the following subsequent paragraph. The power estimates are focused on south Asia and sub-Saharan Africa, as we anticipated that they will account for approximately 50% and 37.5% of

randomized mothers, respectively. The A-PLUS trial is not powered for Latin America because there is only one participating site (Guatemala) and insufficient sample size.

- We have added text to the manuscript to clarify that each study site will aim to enroll an equal number of participants (n=4,250), with no more than 20% of the overall study sample size permitted at any individual site.

Reviewer 2 (Dr. Catherine Oldenburg, University of California San Francisco)

Comments to the Author:

This report is a trial protocol for an exciting and large randomized controlled trial of a single intrapartum 2 g dose of azithromycin versus placebo for reducing maternal and neonatal death or sepsis. The trial is ongoing, and these comments are meant purely for clarification or additional justification. Congratulations to the authors for undertaking this impressive and important trial.

1. Introduction, Line 51: what does “they” refer to here? I assume high risk for infection pregnancies?

RESPONSE: We have revised the text to clarify that “they” refers to pregnancies at high-risk for infection due to prolonged labor or rupture of membranes. This information has been added to the “Introduction” section of the manuscript, specifically in the section entitled “Risk Factors for Maternal and Neonatal Infection and Sepsis”.

2. Introduction Page 4, Line 40: Could the authors provide justification for the dose beyond “seems appropriate”? The previous efficacy in the trial in The Gambia is good rationale; are there other indications for which a 2 g dose is used?

RESPONSE: We have provided further justification for the use of the 2g dose, specifically that the 2g dose is bioequivalent to the 500 mg IV dose used in the US RCT that demonstrated the effectiveness of azithromycin in preventing infection after cesarean delivery. This information has been added to the “Introduction” section of the manuscript, specifically in the section entitled “Rationale for a trial”.

3. Introduction, Page 4, Line 40: The RCT design is the best design to evaluate efficacy, which provides the best evidence for policy. Perhaps consider rewording?

RESPONSE: We have revised the text in the manuscript based on the reviewer’s suggested wording.

4. Methods, Page 11, Line 17: Statistical tests for baseline variables are not considered best practice – see for example <https://pubmed.ncbi.nlm.nih.gov/1967441/>,

<https://pubmed.ncbi.nlm.nih.gov/2171874/> for rationale

RESPONSE: We agree with the reviewer’s comment and have deleted the relevant text from the “Analytical Plan” section.

5. Methods, Page 11, Line 37: Although adherence concerns with single dose azithromycin are often minimal, I would urge caution with an as-treated analysis unless a modern method that does not rely on classic “per protocol” analysis is used – e.g., an instrumental variable approach or similar.

Conditioning on individuals who complied with their randomized treatment assignment can introduce bias. Again, if only a few individuals do not take their randomized treatment assignment, the bias introduced will be minimal if at all, and this is not a concern. But if it is substantial, consider alternative approaches.

RESPONSE: Thank you for this comment. Upon review, we realize that the manuscript did not clearly state that the Intention to Treat population is the primary analysis population. As such, we have added clarifying text to the “Analytical Plan” section of the manuscript. We also appreciate and understand the reviewer’s concerns about as-treated analyses, such as per protocol analysis, and the risk of introducing bias if individuals do not take their randomized treatment assignment. Based on previous experience with drug trials in this population, as well as the dosing methodology (single, one-time dose given immediately after randomization under observation), we anticipate high rates of adherence. However, we have also taken steps to mitigate this risk in our statistical approach. In addition to the per protocol analysis, our Statistical Analysis Plan (SAP) specifies that additional appropriate sensitivity analyses will be done in the event that greater rates of missing data or lower

rates of adherence are observed than the expected. In the interest of brevity, we have opted against adding this text to manuscript but would be happy to do so upon request.

6. Methods, Page 14, Line 5: Why is an alpha of 0.05 used given that there are two co-primary outcomes? Typically, we'd split the P-value for an alpha of 0.025 for each primary outcome. This needs justification.

RESPONSE: The decision to use an alpha of 0.05 was made by the A-PLUS Steering committee (see Table 5 of manuscript). The A-PLUS investigators agreed that although there is only a single intervention and randomization, the outcomes of interest are in two separate populations (i.e., maternal vs. neonatal) and thus appropriate to treat as two independent hypotheses.

7. Although only noted for the subgroup analyses, the authors note they will adjust for “demographic or clinical variables found to differ significantly between treatment arms”. Consider revising or justifying this approach further – first, as noted previously, we don't typically look for statistically significant differences in baseline variables (by definition, differences are due to chance – and with a large sample size such as in this trial, very small differences could be statistically significant). Second, post hoc adjustment that is not pre-specified leaves one open to “fishing”. Adjustment for baseline variables in RCT analyses is certainly acceptable and sometimes preferable (e.g., for baseline variables known to be correlated with the outcome) when pre-specified, but otherwise should be avoided. Given that these are secondary analyses, this is not a major concern, but something for consideration and justification, and any unplanned (post hoc) adjustments should be carefully labeled as such. For discussions on appropriate adjustment in RCT analyses, see for example:

<https://pubmed.ncbi.nlm.nih.gov/1651207/>, <https://pubmed.ncbi.nlm.nih.gov/9620808/>

RESPONSE: We understand and agree with the reviewer's well-articulated and justifiable concerns. No adjustments to demographic or clinical variables are planned for the primary or secondary outcomes. If adjustments are made, it would only be for ancillary/exploratory analyses. To avoid any confusion, we have removed this text from the “Primary Analysis” section of the manuscript.

8. Are there any plans for evaluation of treatment effects or effect modification on the additive scale?

RESPONSE: Thank you for this suggestion. There are no formal plans for evaluation of treatment effects or effect modification at this time but may be considered as an exploratory analysis.

VERSION 2 – REVIEW

REVIEWER	Oldenburg, Catherine University of California San Francisco, Francis I Proctor Foundation
REVIEW RETURNED	22-Apr-2023
GENERAL COMMENTS	Thank you for responding to my comments. Congratulations again on a fantastic study.

VERSION 2 – AUTHOR RESPONSE